# Dynamics of a Bacterial Community in the Anode and Cathode of Microbial Fuel Cells under Sulfadiazine Pressure

**DOI:** 10.3390/ijerph19106253

**Published:** 2022-05-20

**Authors:** Zhenzhen Yang, Hongna Li, Na Li, Muhammad Fahad Sardar, Tingting Song, Hong Zhu, Xuan Xing, Changxiong Zhu

**Affiliations:** 1Institute of Environment and Sustainable Development in Agriculture, Chinese Academy of Agricultural Sciences, Beijing 100081, China; yangzhenz@126.com (Z.Y.); fahadsardar16@yahoo.com (M.F.S.); songtingting0505@163.com (T.S.); zhuchangxiong@caas.cn (C.Z.); 2Department of Engineering Physics, Tsinghua University, Beijing 100084, China; liyana1207@163.com; 3College of Bioscience and Resources Environment, Beijing University of Agriculture, Beijing 100096, China; zhuhong80@bua.edu.cn; 4College of Life and Environmental Science, Minzu University of China, Beijing 100081, China; xingxuanpku@163.com

**Keywords:** air-cathode microbial fuel cell, sulfadiazine, anodic bacteria, cathodic bacteria, synergistic interaction

## Abstract

Microbial fuel cells (MFCs) could achieve the removal of antibiotics and generate power in the meantime, a process in which the bacterial community structure played a key role. Previous work has mainly focused on microbes in the anode, while their role in the cathode was seldomly mentioned. Thus, this study explored the bacterial community of both electrodes in MFCs under sulfadiazine (SDZ) pressure. The results showed that the addition of SDZ had a limited effect on the electrochemical performance, and the maximum output voltage was kept at 0.55 V. As the most abundant phylum, Proteobacteria played an important role in both the anode and cathode. Among them, *Geobacter* (40.30%) worked for power generation, while *Xanthobacter* (11.11%), *Bradyrhizobium* (9.04%), and *Achromobacter* (7.30%) functioned in SDZ removal. Actinobacteria mainly clustered in the cathode, in which *Microbacterium* (9.85%) was responsible for SDZ removal. Bacteroidetes, associated with the degradation of SDZ, showed no significant difference between the anode and cathode. Cathodic and part of anodic bacteria could remove SDZ efficiently in MFCs through synergistic interactions and produce metabolites for exoelectrogenic bacteria. The potential hosts of antibiotic resistance genes (ARGs) presented mainly at the anode, while cathodic bacteria might be responsible for ARGs reduction. This work elucidated the role of microorganisms and their synergistic interaction in MFCs and provided a reference to generate power and remove antibiotics using MFCs.

## 1. Introduction

The misuse and overuse of antibiotics has not only caused environmental pollution, but it has also stimulated the selection of antibiotic resistance genes (ARGs) [1]. Hence, exploring economical and effective treatments to eliminate these pollutants has become a hot topic. 

A microbial fuel cell (MFC) is regarded as a promising alternative treatment to realize waste resource utilization as well as the removal of antibiotics [2,3]. It includes anodic reactions with complex organic compounds as electron donors and cathodic reactions with O_2_, nitrate and nitrite as electron acceptors [2,3]. The mechanism of MFC to remove antibiotics is the combined effect of anaerobic biodegradation and electrical stimulation. Persistent electrical stimulation stimulates the microbial metabolism by transmitting electrons to bacterial cells, and stimulated microorganisms rapidly metabolize antibiotics by secreting enzymes [4]. Antibiotics, including sulfadiazine (SDZ), sulfamethoxazole (SMX), Tetracycline (TC), Oxytetracycline (OTC), and Chloramphenicol (CAP), could be removed in two dual chamber MFCs [4,5,6], and the voltage output was virtually not affected. Different from two dual chamber MFCs, single-chamber MFCs (mainly air-cathode MFCs) could directly use O_2_ in the air as an electron acceptor without the cathode chamber, so they could increase the mass transfer effective and reduced cost, and O_2_ diffusion from air to the cathode would help the removal of nitrogen without aeration. Therefore, air-cathode MFCs also showed excellent performance in removing antibiotics [4,7,8].

Bacterial community structure played a key role during the power output and contaminant removal in MFCs. Therefore, it is necessary to understand the microbial status in the MFCs to clarify the relationship between power generation and pollutant removal. Previous research revealed that the dominant phylum in MFCs were Proteobacteria, Bacteroidetes, Firmicutes, and Actinobacteria. The bacteria affiliated with these phyla were classified into two groups (degradation related and exoelectrogenic) [9]. The mechanisms of electroactive microorganisms have been summarized in several recent reviews [10,11,12]. These cases of research mainly focused on anodic bacteria, but the growth of bacteria in the cathode is inevitable for air-cathode MFCs, which was an important factor reducing the electrochemical performance [13,14]. Conversely, cathode-biofilm also could prevent the diffusion of O_2_ in the cathode side to the anode chamber to increase the power generation of MFCs [15]. Therefore, the role of cathodic bacteria is controversial. There is limited literature on cathodic bacteria of air-cathode MFCs. To gain insight into anodic and cathodic communities, Daghio et al. firstly operated a single chamber MFC to investigate microbial communities. It was found that degradation-related bacteria were enriched in the cathode, but electricigens or closely related microbes were clustered in the anode to attribute chlorinated herbicide removal or power generation in soil MFCs [16,17]. Yuan et al. found that the COD/N of wastewater would affect the bacteria, both in the anode and cathode of MFCs [18]. The distribution trend of nitrifiers and denitrifiers in cathodes varied with the cathode-biofilm depth. These studies demonstrated that the cathode played an important role in power generation and contaminants removal in air-cathode MFCs; however, the research on the degradation of antibiotics in wastewater by MFCs mostly focused on anode biofilm, and the function of cathodic bacteria was seldom mentioned. Therefore, it is necessary to clarify the roles of both the anode and cathode in power generation, antibiotic degradation, and ARG propagation.

This work investigated the efficacy of air-cathode MFCs in wastewater treatment, with SDZ as the representative antibiotic. SDZ is reported to be one of the most common sulfonamides and is used frequently in veterinary medicine [19]. The electrochemical and physicochemical performance of MFCs under SDZ pressure was studied. Especially, the structures and interactions between anodic and cathodic microorganisms were mainly analyzed. Finally, the occurrence of ARGs and integrons in the anode and the cathode are discussed. The study will elucidate the role of microorganisms in both electrodes and provide reference for the further application of air-cathode MFCs in pollution control and power generation.

## 2. Materials and Methods

### 2.1. Chemicals and Reagents

Analytical reagent sulfadiazine was purchased from Biotopped Science & Technology Co., Ltd. (Shanghai, China). A graphite fiber brush and carbon cloth were acquired from Cetech CO., Ltd. (Suzhou, China). The catalyst (20% Pt/C) was obtained from Johnson Matthey Co., Ltd. (Shanghai, China). The concentration of stock solution for sulfadiazine was 5 g/L and preserved in a 4 °C refrigerator for further use. Other chemicals were purchased from Sinopharm chemical regent Co., Ltd. (Shanghai, China).

### 2.2. MFC Start-Up and Operation

The MFC reactors were constructed with a fiber brush anode and carbon cloth cathode (Appendix A). They were inoculated with granule sludges collected from a wastewater plant in Shangdong, China. The composition of the anolyte is shown in Appendix A. The external resistance was fixed at 1000 Ω, and titanium wire (Φ 1 mm) was used as wires connected to the carbon cloth. The reactors were stabilized for two weeks in a biochemical incubator at 35 °C. The medium was changed every 48 h during acclimation; After that, the reactors were operated with the sequencing batch, and the medium was changed every 24 h. The whole experiment was carried out at ambient temperature in the dark. The reactors were designed MFCs and open-circuit control with three replicates of each. A total of 40 mg/L of SDZ was added in each reactor after the output voltage reached stabilization. 

### 2.3. Analytical Methods

#### 2.3.1. Determination of Physicochemical Parameters and SDZ

Because there were many suspended solids in the MFCs effluent, it was easy to cause deviation due to uneven sampling. The samples were filtrated by a 0.45-μm filter (Tianjin jinteng experimental equipment Co., Ltd., Tianjin, China) and detected. The concentrations of chemical oxygen demand (COD), ammonium-nitrogen (NH_4_^+^-N), and total dissolved nitrogen (TDN) were analyzed according to the standard method (See Appendix A). The pH and conductivity were measured with a pH-meter and conductivity gauge. The concentration of SDZ was detected by high-performance liquid chromatography (Ultimate 3000, Thermofisher, Waltham, MA, USA) after filtering by a syringe filter (0.22 μm), The samples were collected by an automatic sampling device and injected into C18 liquid chromatography (0.46 × 25 cm, 5 μm, Thermofisher, Waltham, MA, USA). During the elution of the mobile phase, a UV detector was used for analysis and detection at 270 nm. The mobile phase comprised acetonitrile/3‰ acetic acid (*v*/*v* = 25/75), the flow rate was 1 mL/min, and the injection volume was 20 μL.

#### 2.3.2. Electrochemical Measurement

The external voltages were collected using a data acquisition board (PISO-813, ICP DAS Co., Ltd., Shanghai, China) via online monitoring and recording every 30 min. The power density was normalized by the projected surface area of one side of the cathode. A standard three-electrode system was employed for electrochemical tests. The anode was the working electrode, while the cathode was the counter electrode, and Ag/AgCl was used as the reference electrode. Cyclic voltammetry (CV) was carried out on the electrochemical workstation (CHI660E, CH Instrument Ins.) by sweeping the cell potential from −0.8 V to +0.6 V at a rate of 5 mV/s. Electrochemical impedance spectroscopy (EIS) measurements were conducted at the open circuit voltage (OCV) of the MFCs with the amplitude set to 5 mV and using a frequency range of 100 kHz–0.1 Hz, with the anode chamber filled with a fresh anolyte (Appendix A).

#### 2.3.3. DNA Extraction and qPCR of ARGs

The samples were collected from the MFCs anode and cathode biofilms in triplicate. The shape of the carbon brush and carbon cloth were different, and for the convenience of the comparison, weight was used for quantification (1/4 of carbon brush and 1/6 of carbon cloth was used for DNA extraction). DNA extraction and real-time quantitative PCR (qPCR) were performed according to methods used in a previous study [20]. DNA samples were stored at −20 °C until use. Each qPCR was repeated three times. The genes of 16S rRNA, *intI*1, *intI*2, *sul*1, *sul*2, *sul*3, and *sul*A were tested, and the primers were set in Appendix A.

#### 2.3.4. Bacterial Community Analysis

High throughput sequencing was performed at Beijing Allwegene Technology Co. Ltd., Beijing, China. See Appendix A for details.

#### 2.3.5. Data Analysis

The basic statistical calculations and analysis were performed using SPSS 23.0 (IBM, Chicago, IL, USA) and Origin 9.1 (Origin Lab, San Diego, CA, USA). *p* < 0.05 was considered statistically significant. The changes of the bacterial community at the phylum-level made use of Circos-0.67–7 software (http://circos.ca/, accessed on 8 March 2021), and the networks were performed using Networkx software (http://networkx.github.io/, accessed on 20 August 2021) (*p* < 0.05), according to the relative contents of each genu after classification.

## 3. Results and Discussion

### 3.1. Performance of Air-Cathode MFCs under Sulfadiazine Pressure

The removal ratio of COD, NH_4_^+^-N, and TDN was 86.55%, 45.15%, and 45.64% in the MFCs, respectively, and it was 83.46%, 9.56%, and 11.27% in the open circuit treatment, respectively (Figure 1a). The degradation of SDZ was gradually accelerated with the process (Figure 1b). Twelve cycles later, over 50% of SDZ could be removed within a cycle, which suggests that the microbes in the MFCs gradually acquired the ability for SDZ degradation. Compared with the open circuit, MFCs reduced the increase of pH and conductivity (Appendix A) to keep a suitable environment for microorganisms to degrade pollutants, even under SDZ pressure. In air-cathode MFCs, oxygen diffused from the air into the biofilm from the cathode, which played a role in nitrification and mono-oxygenation reactions for the removal of SDZ [21,22]. Compared with other research (13.39–80%) [6,23], our experiment was conducted in the dark, which might have caused a relatively low removal ratio of SDZ (58.72%).

The maximum output voltage was 0.55 V (Figure 1c), and the addition of SDZ had no drastic effect on the output voltage of MFCs. Previous studies have demonstrated that the output of MFCs would be enhanced rather than be inhibited under antibiotic pressure [7,8,24]. An obvious peak was at −0.4 V vs. Ag/AgCl in CV curves (Figure 1d), which was close to the formal potential of outer membrane cytochromes of *G. sulfurreducens* (−0.398 V vs. Ag/AgCl) [25]. It was consistent with a previous study that concluded that some antibiotics could increase the permeability of exoelectrogens membranes and then facilitate the direct electron transfer [7]. In addition, the maximum output power (Appendix A) and Nyquist plots (Appendix A) demonstrate that the addition of SDZ showed little influence on the electrochemical activity of MFCs.

### 3.2. Bacterial Community Shift in the Anode and Cathode under Sulfadiazine Pressure

The goods coverage of all tested samples was 99%, which indicated that the sizes of all libraries were enough to cover the bacterial communities. As shown in Figure 2, the bacterial community showed a lower Chao1 index and a higher Shannon index in the cathode compared with that of the anode. This indicated that although the bacteria community in the cathode was not rich, it had a higher diversity. The principal coordinate analysis (PCoA) based on UniFrac distances showed that the sample had a good reproducibility (Appendix A). The distance between samples further validated the difference of the bacterial community structures between the anode and cathode. The unique species also sharply reduced in the anode and cathode (Appendix A), which will be discussed in detail below. These indicated that the bacteria in the inocula were selectively enriched in the anode and cathode.

Firmicutes, Ignavibacteriae, Chloroflexi, Lentisphaerae, and Euryarchaeota were mainly enriched in the anode, while Actinobacteria mainly clustered in the cathode; Proteobacteria and Bacteroidetes existed both in the anode and cathode (Figure 3). Among these phyla, Proteobacteria, Bacteroidetes, and Actinobacteria were three dominant phyla with the relative abundance of 88.6% (anode) and 97.1% (cathode), respectively. The function of these bacteria determined their selective enrichment in the anode and cathode, which will be discussed in more detail at follow-up.

Proteobacteria was most abundant in the anode (70.7%) and cathode (62.6%). The difference was further explicated at the class-level (Figure 4a). The most abundant class in anodic community was Deltaproteobacteria (40.91%), for which almost all the detected sequences belonged to the genus *Geobacter* (Figure 4b). *Geobacter* was the dominant genus in the MFCs anode, with the relative abundance of 40.3% as Gram-negative and electroactive bacteria [26,27]. *Geobacter* is strictly anaerobic and could adapt to the low surface potentials of the anode [26,27]; therefore, they mainly colonized in the anode and were responsible for power generation by using acetate. *Xanthobacter* (11.11%, *p* < 0.01), *Bradyrhizobium* (9.04%, *p* < 0.01), and *Mesorhizobium* (2.68%, *p* < 0.001) affiliated with Alphaproteobacteria were enriched in the cathode (Figure 4b). *Xanthobacter* was dominant in the microbial electrolysis cells (MECs) but was scarcely reported in MFCs. They accumulated in the cathodic community in this experiment, possibly due to the fact that *Xanthobacter* thrives in micro-oxygen environments [18], as well as that they could degrade the aromatic structure of SDZ in the cathode [28]. Similarly, *Bradyrhizobium* could degrade antibiotics via co-metabolism with acetate [29]. *Mesorhizobium* is an aerobic bacterium using oxygen as the terminal electron acceptor to respire. The possible role of *Mesorhizobium* was to remove acetate and SDZ degradation products because it could metabolize amino salts, nitrates, and various amino acids using various carbohydrates and organic acid as the carbon source [30]. *Azospirillum* (1.88%, *p* < 0.01) affiliated with Alphaproteobacteria was enriched in the anode, which have been reported to be resistant to antibiotics using a wide range of carbon sources [31]. It was reported that *Bradyrhizobium*, *Mesorhizobium*, and *Azospirillum* could produce extracellular polysaccharides, which supported their metabolism [30,31]. *Achromobacter* (7.30%, *p* < 0.001) and *Castellaniella* (2.24%, *p* < 0.01) affiliated with Betaproteobacteria were enriched in the cathode (Figure 4b). *Achromobacter* was able to degrade sulfonamides [32], and *Castellaniella* could degrade pyrene under denitrifying conditions [33], so they clustered in the cathode to degrade the aromatic structure of SDZ. *Azospira* (5.45%, *p* < 0.01) affiliated with Betaproteobacteria was enriched in the anode as denitrification organisms, which could denitrify under anaerobic conditions and could use the electrode as the electron acceptor [34]. They colonized in the anode for denitrification and power generation. *Hydrogenophaga*, an autotrophic H_2_-oxidizing bacteria, could utilize hydrogen as the energy source [35]. *Comamonas* as facultative anaerobic microorganisms have been isolated from MFCs, which contributed to the power generation with acetate as the electron donor [36]. They belong to Betaproteobacteria, with lower abundance in the anode and the cathode (*Hydrogenophaga* (0.49% vs. 2.28%, *p* = 0.114) and *Comamonas* (0.78% vs. 0.05%, *p* = 0.403)). *Pseudomonas* (4.58%, *p* < 0.01), which belong to Gammaproteobacteria, were enriched in the anode (Figure 4b). As electroactive bacteria, *Pseudomonas* were also able to degrade sulfonamides, which were the dominant bacterial genus in the anodic chamber of MFCs [37], so they accumulated in the anode for SDZ removal and power generation. *Stenotrophomonas* (1.59% vs. 2.68%, *p* = 0.186) and *Dokdonella* (1.20% vs. 2.18%, *p* < 0.01) affiliated with Gammaproteobacteria were enriched in both the anode and cathode. *Stenotrophomonas* is able to degrade many xenobiotic aromatic compounds [38]. *Dokdonella* is usually found in the aerobic biological systems to remove nitrogen and degrade aromatic hydrocarbons simultaneously [39], so they colonized both the anode and cathode, possibly for the degradation of SDZ.

Actinobacteria were widely used in water treatment field, as they can use glucose, starch, and cellulose as carbon sources [40]. The abundance of Actinobacteria in the cathode (19.68%) was higher than that in the anode (7.65%, *p* < 0.01) (Figure 4a), due to the anaerobic condition that inhibited Actinobacteria [41]. *Microbacterium* (9.85%, *p* < 0.05) and *Pseudoclavibacter* (2.78%, *p* < 0.001), which belong to Actinobacteria, clustered in the cathode (Figure 4c). *Microbacterium* is an important degradation-related microorganism, which showed a higher degradation rate for sulfonamides, especially for SDZ [42]; therefore, it was most likely responsible for the degradation of SDZ in the cathode. *Pseudoclavibacter* is a Gram-positive and aerobic genus; although this genus has not been reported in the wastewater treatment field, it was reported to be responsible for the biodesulfurization of organic sulfur [43]; therefore, they might cluster in the cathode, helping the removal of sulfur in SDZ. *Rhodococcus* (2.69% vs. 1.99%), *Mycobacterium* (1.03% vs. 0.43%), and *Gordonia* (1.56% vs. 3.69%, *p* < 0.01) affiliated with Actinobacteria existed both in the anode and the cathode (Figure 4c). *Rhodococcus* degraded SDZ and regulated biofilm thickness in both the anode and cathode, for they could degrade various aromatic compounds via a ring-cleavage pathway under anaerobic conditions [44]. They could also improve MFC performance via controlling the biofilm thickness on the anode surface [45]. Mycobacterium and *Gordonia* were beneficial to degrade SDZ, for they could degrade polycyclic aromatic hydrocarbons (PAHs) into phthalate, CO_2_, and other chemicals by decarboxylation, dioxygenation, hydrolysis, and ring-cleavage reactions [6,46]. *Gordonia* is an aerobic bacterial genus, but it was found in the anode in previous studies [46,47], which is possibly due to less attention being paid to microorganisms in the cathode of MFCs.

Bacteroidetes were enriched both in the anode (10.28%) and cathode (14.86%) without a significant difference between each other. *Proteiniphilum* (2.02%, *p* < 0.05) and *Petrimonas* (1.83%, *p* < 0.05) affiliated with Bacteroidetes mainly accumulated in the anode (Figure 4c), possibly because they are typical fermenters. *Proteiniphilum* and *Petrimonas* could degrade complex substrates, such as PAHs, to simple organic compounds by syntrophic metabolism [48,49], which indicates that they could degrade the product of SDZ in anode. There was no significant difference for the relative abundance of *Chryseobacterium* between the anode (1.27%) and the cathode (1.17%) (Figure 4c). Considering their ability to degrade aromatic compounds [50], they might degrade SDZ in both electrodes.

Generally, Proteobacteria played an important role both in the anode and cathode, without a significant difference as the predominant phylum: Deltaproteobacteria mainly accumulated in the anode, represented by the genus of *Geobacter*; it was responsible for power generation. Alphaproteobacteria, Betaproteobacteria, and Gammaproteobacteria colonized both the anode and cathode and were associated with the degradation of SDZ; Actinobacteria mainly clustered in the cathode, which were responsible for the removal of SDZ; Bacteroidetes showed no significant difference between the anode and cathode, and they were associated with the degradation of SDZ. The nature (aerobic/anaerobic) and function of bacterial genera decided their distribution in the anode and cathode.

### 3.3. Potential Bacterial Roles in SDZ Degradation

The microorganisms in air-cathode MFCs can be functionally categorized into the following two groups: exoelectrogenic bacteria (*Geobacter*, *Pseudomonas*, *Azospira*, and *Comamonas*) and degradation-related bacteria (*Xanthobacter*, *Bradyrhizobium*, *Achromobacter*, *Azospirillum*, *Microbacterium*, *Pseudoclavibacter*, *Mycobacterium*, *Castellaniella*, *Dokdonella*, *Rhodococcus*, *Mycobacterium*, *Gordonia*, *Proteiniphilum*, *Petrimonas*, and *Chryseobacterium*). Based on the variation of bacterial communities in the biofilms, the ecological model of SDZ biodegradation in air-cathode MFCs was proposed (Figure 5a). Substrates (acetate and SDZ) diffused into the biofilm both in the anode and cathode, while oxygen diffused from the cathode side. In the anodic biofilms, *Geobacter* was mainly responsible for power generation by the metabolizing acetate, and SDZ degradation products were further oxidized by fermentation bacteria (e.g., *Proteiniphilun* and *Petrimonas*) or *Gordonia* to low molecular organics. Nitrification bacteria used the O_2_ from cathode. The oxygen-utilizing of cathodic bacteria can create anoxic zones, offering a favorable zone for denitrifying bacteria and exoelectrogenic bacteria. Almost all cathodic bacteria and part of anodic bacteria were involved in nitrification and denitrification. 

Based on published literature, the possible degradation pathway of SDZ is shown in Figure 5b. Furthermore, the role of microorganisms in the degradation of SDZ and their relationship in air-cathode MFCs was explained according to specific functions. It is noteworthy that the initial electrophilic attack by oxygenases of aerobic bacteria is often a rate-limiting step and the first of a chain of reactions which is responsible for the biodegradation of many organic compounds [52]. This might be the main reason why degradation bacteria were mostly distributed in the cathode. In the pathway I, SDZ obtained electrons in the presence of O_2_ to form dihydroxyl SDZ (I-①). As mentioned in Section 3.2, *Mycobacterium* might work in the process by deoxygenation. Moreover, *Microbacterium* was able to degrade SDZ into 2-aminopyrimidine, which was initiated by NADH-dependent ipso-hydroxylation [53], corresponding to the S-N bond hydrolysis, (I-②) or (II-①). The *p*-anilinesulfonic acid was further hydrolyzed to generate aniline and sulfate (I-③), or N atoms were bio-consumed to form benzenesulfinic acid (II-②), then leading to the formation of benzene and sulfate. During this process, *Pseudoclavibacter* was responsible for biodesulfurization. The benzene and catechol can be decomposed into acetate and pyruvate by *Proteiniphilun*, *Petrimonas*, *Rhodococcus*, and *Castellaniella* (I-⑥, II-④). 2-amino-4,6-dihydro-pyrimidine contains more electron-donating functional groups, which render the molecules more prone to electrophilic attack by oxygenases of aerobic bacteria (mainly cathodic bacteria); then, it was decomposed into N_2_, formic acid, and acetate by nitrifiers/denitrifiers (I-⑦, II-⑥) [52]. Finally, formic acid, acetate, and pyruvate can be used by a *Geobacter* for power generation. The degradation of SDZ was the result of the synergistic reaction of anodic and cathode bacteria in air-cathode MFCs. O_2_ was harmful to power generation in MFCs, but it was conducive to remove contaminants. Therefore, the role of O_2_ in air-cathode MFCs should be reconsidered.

All the reactions in air-cathode MFCs can be summarized as follows: (1) Exoelectrogenic bacteria were responsible for power generation in the anode of air-cathode MFCs; (2) The electrons provided by exoelectrogenic bacteria could increase the metabolic reaction of degradation-related bacteria located in the anode and cathode; (3) Degradation-related bacteria might contribute to the power generation by producing metabolites, which could be used as electron donors by the electroactive bacteria. These functional bacteria could effectively remove pollutants and generate power via complex synergistic interactions.

### 3.4. ARGs in the Anode and Cathode under Sulfadiazine Pressure

As shown in Figure 6a, 16S rRNA gene, *intI*1, *intI*2, *sul*1, and *sul*2 were detected in the anode and cathode. This is consistent with previous work that showed that *sul*3 and *sul*A were not detected in any samples [54]. The integrons, *intI*1 and *intI*2, are natural mobile genetic elements that can capture, integrate, and express resistance gene cassettes with the help of integrase genes. Integrase genes, being important players in ARG transfer, were the driving force for bacterial evolution. Integrons have been reported to be involved in the occurrence of new resistant and pathogenic species [8]. The resistance of bacteria to sulfonamides are mutated of the enzyme dihydropteroate synthase (DHPS) or acquired alternative DHPS gene, among which sul1 and *sul*2 belong to the latter [55]. The absolute abundance of 16S rRNA gene, *intI*1, *intI*2, *sul*1, and *sul*2 in the anode was higher than that of the cathode (Figure 6a), and a significant difference was observed between the anode and cathode. Moreover, the relative abundance of these ARGs in the anode was also higher than that in the cathode, except for *sul*2 (Figure 6b). The relative abundance of *sul*1 and *sul*2 ranged from 6.59 × 10^−4^ to 3.43 × 10^−2^, and *sul*2 > *sul*1 in the cathode. Similar to other studies, the abundance of *sul*1 was usually lower than *sul*2 [56].

A network analysis was conducted in order to further explore the relationship of the bacterial community, integrons, and ARGs in the air-cathode MFCs (Figure 6c). *intI*1 showed a positive correlation with *Geobacter* and presented a negative correlation with *Bradyrhizobium*, *Mesorhizobium*, *Achromobacter*, and *Hydrogenophaga*. It indicated that *Geobacter* might be the host bacteria of *intI*1, while the proliferation of others might be effective to reduce *intI*1. *Geobacter* mainly clustered in the anode, and others were enriched in the cathode, which explains why the abundance of *intI*1 in the anode was higher than that in the cathode. The *sul*1 presented a positive correlation with *Proteiniphilun*, *Petrimonas*, and *Azospirillum* and showed a negative correlation with *Microbacterium*, *Dokdonella*, *Stenotrophomonas*, and *Castellaniella*. *Proteiniphilun*, *Petrimonas*, and *Azospirillum* were mainly enriched in the anode, and *Microbacterium* and *Castellaniella* mainly clustered in the cathode. Though *Stenotrophomonas* and *Dokdonella* were both enriched in anode and cathode, the abundance of these bacteria in the cathode was higher than that in the anode. All of these were consistent with the abundance of *sul*1 in the anode being higher in the cathode. The *sul*2 showed a positive correlation with *Hydrogenophaga* and a negative correlation with *Geobacter*. In addition, the relative abundance of *Geobacter* in the anode was substantially higher than the abundance of *Hydrogenophaga* in the cathode. This was consistent with the relative abundance of *sul*2 being lower in the anode than that of the cathode. The results showed that many anodic bacteria were potential hosts of the tested ARGs, while the cathodic bacteria might play a role in the reduction of these ARGs. The findings were consistent with previous research that aerobic conditions showed better removal capacities of ARGs than anaerobic conditions [57].

## 4. Conclusions

In summary, the addition of SDZ has a limited effect on the electrochemical performance of air-cathode MFCs, with the maximum output voltage kept at 0.55 V. Anodic bacteria were mainly responsible for power generation, and part of them could remove contaminants. Cathodic bacteria were responsible for pollutants removal. The nature (aerobic/anaerobic) and function of bacteria decided their distribution in the anode and cathode. The electrons provided by exoelectrogenic bacteria could increase the metabolic reaction of degradation-related bacteria, and degradation-related bacteria might contribute to the power generation by producing secondary metabolites. They could remove SDZ and generate power through synergistic interactions. The potential hosts of ARGs mainly presented in anodic bacteria, while cathodic bacteria possibly played a role in ARG reduction. Regardless, the spread risk of antibiotic resistance should be seriously concerned for both electrodes.

## Figures and Tables

**Figure 1 ijerph-19-06253-f001:**
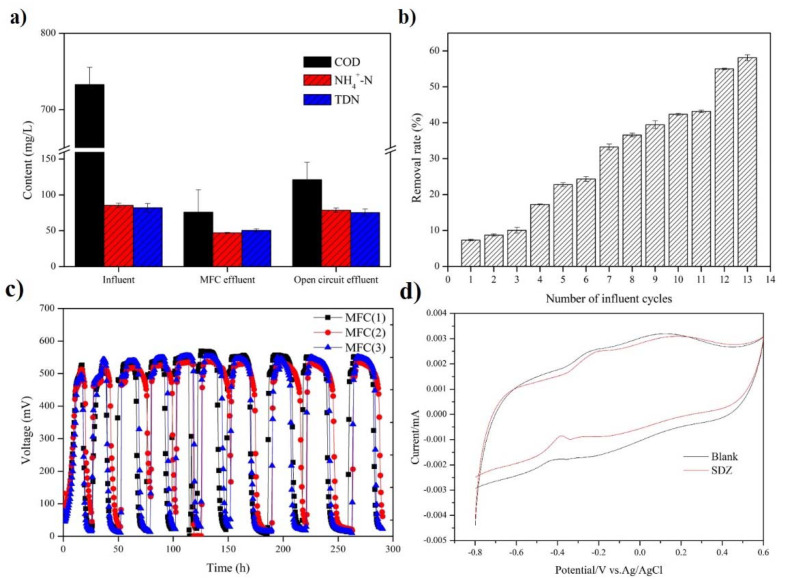
Performance of the air-cathode system: (**a**) Contents of COD, NH_4_^+^-N, and DTN in MFCs; (**b**) Removal rate of SDZ; (**c**) Output voltage; (**d**) CV curves.

**Figure 2 ijerph-19-06253-f002:**
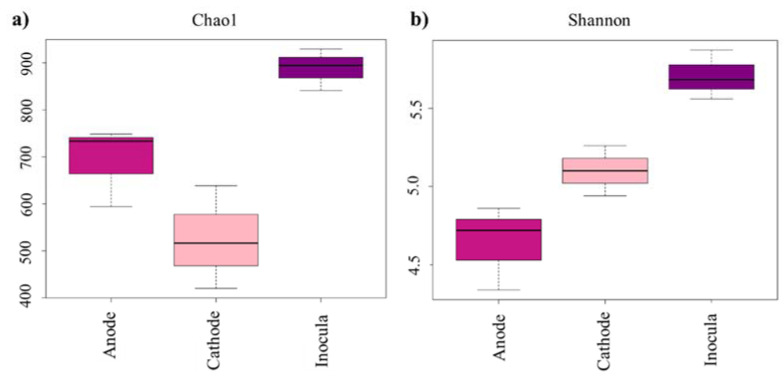
The microbial diversity of samples: (**a**) Chao1 index; (**b**) Shannon index.

**Figure 3 ijerph-19-06253-f003:**
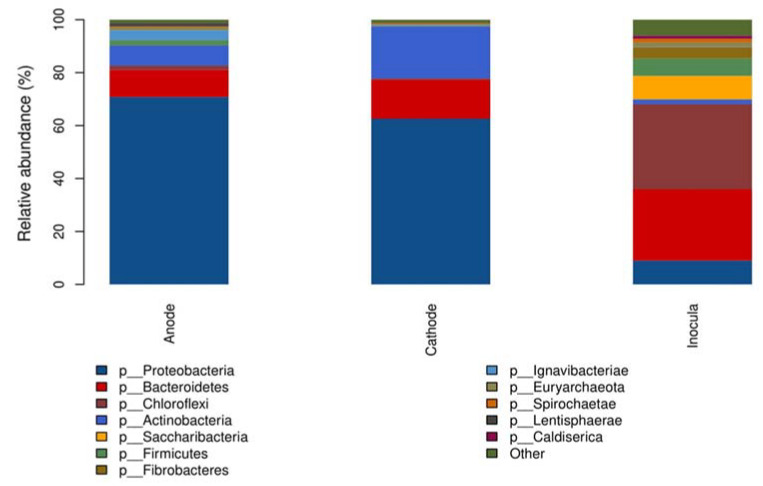
Phylum-level microbial communities.

**Figure 4 ijerph-19-06253-f004:**
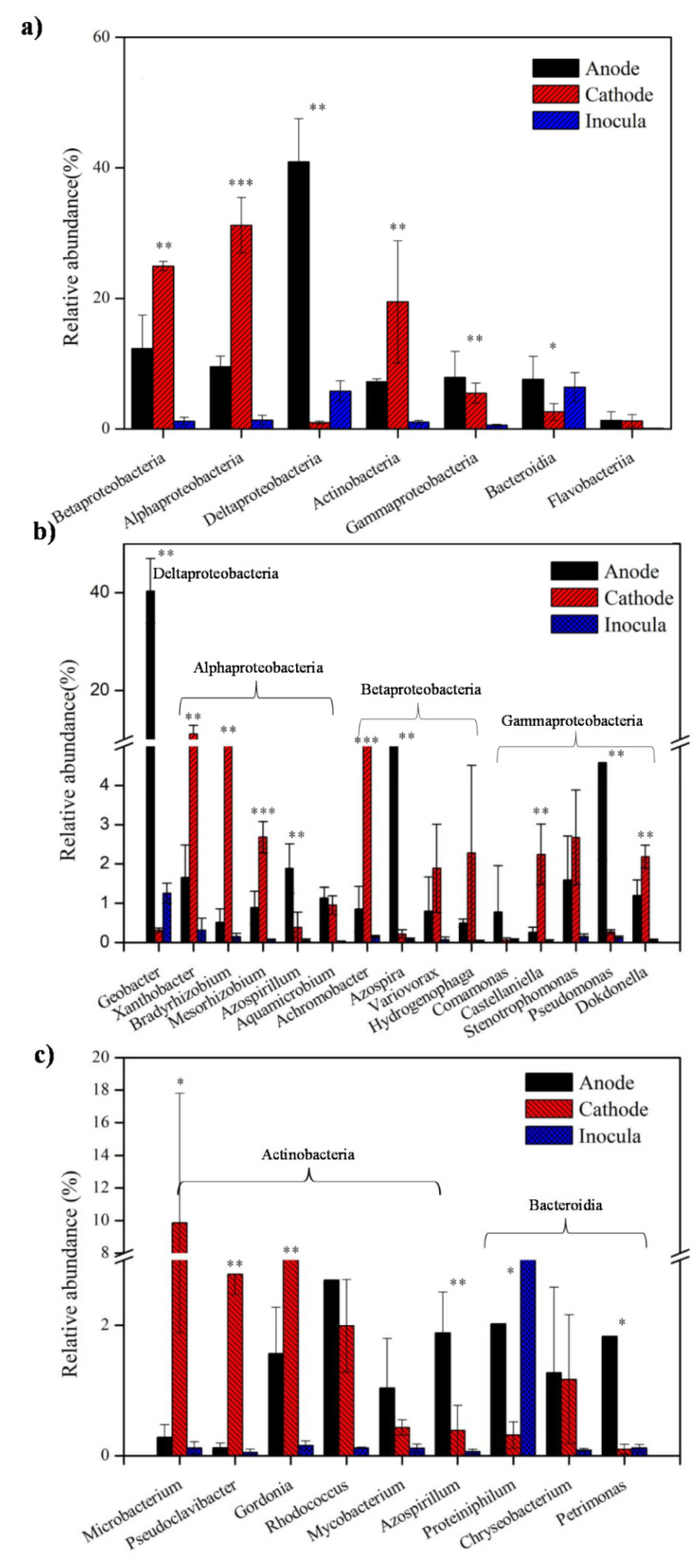
The relative abundance of (**a**) class-level microbial communities, (**b**) genera belonging to Proteobacteria, and (**c**) genera belonging to Actinobacteria and Bacteroidetes (Analysis of difference between anode and cathode: * *p* < 0.05, ** *p* < 0.01, *** *p* < 0.001).

**Figure 5 ijerph-19-06253-f005:**
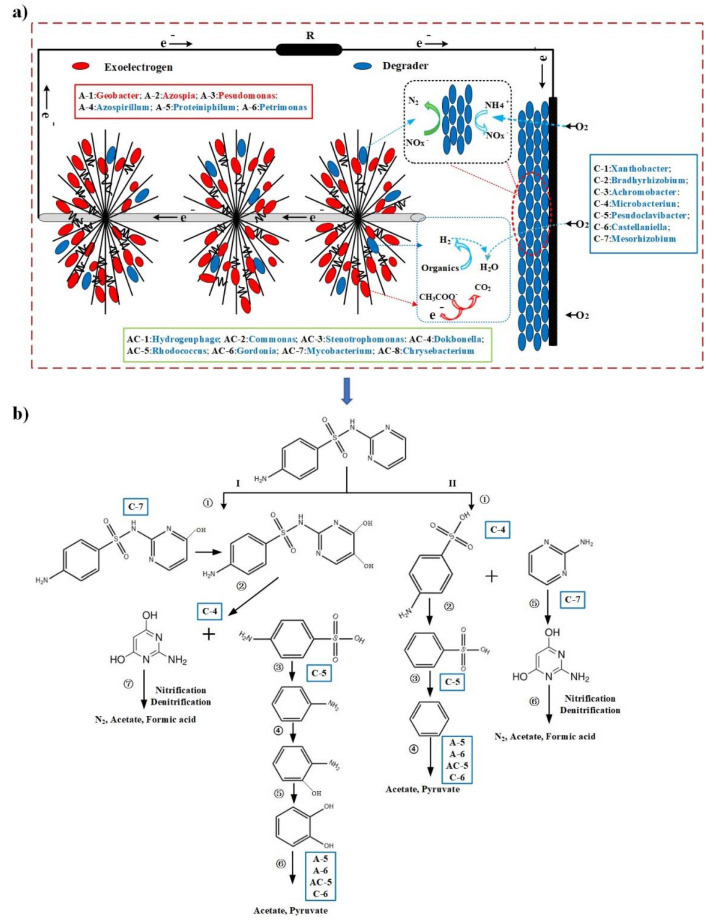
(**a**) Ecological model of SDZ biodegradation in air-cathode MFCs; (**b**) Scheme of potential degradation mechanisms [6,42,51]. (A-n: the genus mainly accumulated in the anode; C-n: the genus mainly clustered in the cathode; AC-n: the genus clustered both in the anode and cathode; I and II: Possible SDZ biodegradation pathway).

**Figure 6 ijerph-19-06253-f006:**
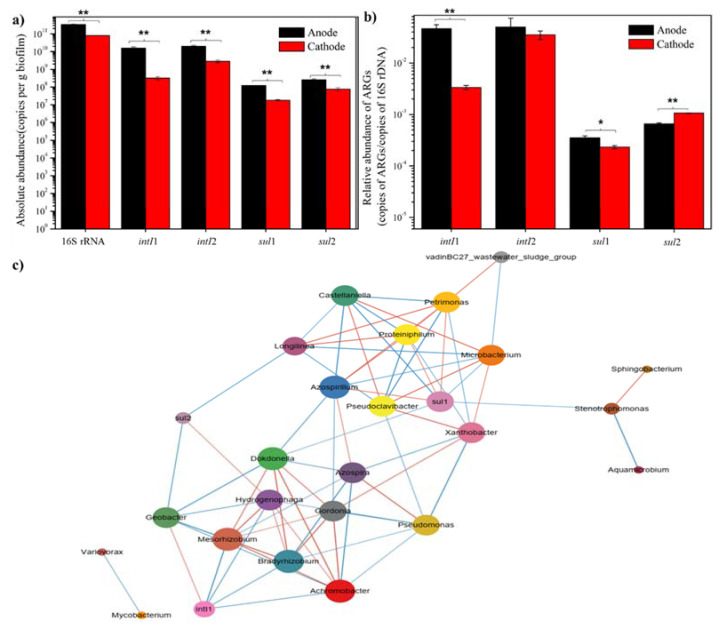
(**a**) Absolute abundance of ARGs; (**b**) Relative abundance of ARGs. (Analysis of difference between anode and cathode: * *p* < 0.05, ** *p* < 0.01); (**c**) Network analysis of ARGs and the bacterial community in MFCs. Species with *p* < 0.05 were shown by default according to Spearman’s rank analysis. Node size was weighted according to the degree. The orange color represents the positive correlation, and the blue represents the negative correlation.

## Data Availability

Not applicable.

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
