# Peer review of "Dynamics of a Bacterial Community in the Anode and Cathode of Microbial Fuel Cells under Sulfadiazine Pressure"

_ijerph, 2022, doi:10.3390/ijerph19106253_

Round 1

Reviewer 1 Report

The present article deals with Microbial Fuel Cells to assist in the degradation of an antibiotic, sulfadiazine. The main aspect was to investigate the effects of sulfadiazine on the MFC operation, as well as to study the microbial community on the electrodes. 

Overall, the article is very well written and seems to be scientifically sound. Therefore, only minor adjustments to the manuscript and the supplementary material such as wording or formatting should be considered. Some improvements to the graphics are also advisable. All suggested corrections have been highlighted in the attached files.

Furthermore, the statistical methods could be described a bit more in detail, as they are a central point of this article.

Good luck with further revision.

Author Response

Manuscript No.: ijerph-1711824

Title: Dynamics of Bacterial Community in the Anode and Cathode of Microbial Fuel Cells Under Sulfadiazine Pressure

Response to editor:

Dear Dr. Wu,

Thank you for considering our manuscript titled “Dynamics of Bacterial Community in the Anode and Cathode of Microbial Fuel Cells Under Sulfadiazine Pressure”. Please note that we have made corrections of our manuscript based on the comments. During this process, we have carefully considered all issues mentioned in the reviewers' comments, and the issues were marked up using the “Track Changes” function. In addition, we have our manuscript polished English by a native speaker.

Now we’d like to re-submit our manuscript and we look forward to your positive consideration.

Yours sincerely,

Hongna Li

Institute of Environment and Sustainable Development in Agriculture, Chinese Academy of Agricultural Sciences, Beijing 100081, PR China

Response to Reviewer:

Reviewer:

The present article deals with Microbial Fuel Cells to assist in the degradation of an antibiotic, sulfadiazine. The main aspect was to investigate the effects of sulfadiazine on the MFC operation, as well as to study the microbial community on the electrodes.

Overall, the article is very well written and seems to be scientifically sound. Therefore, only minor adjustments to the manuscript and the supplementary material such as wording or formatting should be considered. Some improvements to the graphics are also advisable. All suggested corrections have been highlighted in the attached files.

Furthermore, the statistical methods could be described a bit more in detail, as they are a central point of this article.

Response: Thank you very much for your kind review and comments. We have modified the issues in the attached files made by you, and corrected as much as we could in our manuscript according to your comments. We look forward to your positive consideration for the revised manuscript.

Specific comments:

Comment:The statistical methods could be described a bit more in detail, as they are a central point of this article.

Response: Thank you for your comment, we have described the statistical methods in section 2.3.5 Data analysis. (Line 145-151)

Basic statistical calculations and analysis were performed using SPSS 23.0 (IBM, Chicago, Ill, USA) and Origin 9.1 (Origin Lab, San Diego, CA, USA). p < 0.05 was considered statistically significant. The changes of bacterial community at phylum level made use of Circos-0.67–7 software (http://circos.ca/). and the networks were performed using Networkx software (http://networkx.github.io/) (p < 0.05) according to the relative contents of each genu after classification.

Reviewer 2 Report

The present manuscript shows the dynamics of the bacterial community in the anode and cathode of microbial fuel cells under the addition of the model antibiotic sulfadiazine.

The authors have successfully assessed the microbial structure of anodic and Cathodic biofilms in heir experiments.

They have also successfully quantified at which extent theirs systems have removed the model antibiotic.

What it is not completely clear is the function of the microorganisms within the Cathodic biofilm.

Given the composition and the electron donor of the medium in the anodic chamber, I can clearly see that very likely the electroactive microorganisms within the biofilm were responsible of acetate oxidation.

On the cathode side, it is not clear the function of microorganisms in contact with the cathode material. This requires a detailed explanation.

According to the available information:

-What could be the electron transfer mechanism by microorganisms at the cathode?

-Transferred electrons from anode to cathode are used to abiotically/biotically reduce O2 into something else?

Additionally, precise details should be given to clarify the experimental conditions at which CV and EIS were performed:

-Were anodic, cathodic or both biofilms subjected to electrochemical analysis?

-Were biofilms subjected to analysis under the presence of substrate, at the maximum of current production or under complete absence of substrate.

This requires clarification.

Author Response

Response to Reviewer:

Reviewer:

The present manuscript shows the dynamics of the bacterial community in the anode and cathode of microbial fuel cells under the addition of the model antibiotic sulfadiazine.

The authors have successfully assessed the microbial structure of anodic and Cathodic biofilms in their experiments.

They have also successfully quantified at which extent their systems have removed the model antibiotic.

What it is not completely clear is the function of the microorganisms within the Cathodic biofilm.

Given the composition and the electron donor of the medium in the anodic chamber, I can clearly see that very likely the electroactive microorganisms within the biofilm were responsible of acetate oxidation.

On the cathode side, it is not clear the function of microorganisms in contact with the cathode material. This requires a detailed explanation

According to the available information:

-What could be the electron transfer mechanism by microorganisms at the cathode?

-Transferred electrons from anode to cathode are used to abiotically/biotically reduce O2 into something else?

Additionally, precise details should be given to clarify the experimental conditions at which CV and EIS were performed

-Were anodic, cathodic or both biofilms subjected to electrochemical analysis?

-Were biofilms subjected to analysis under the presence of substrate, at the maximum of current production or under complete absence of substrate.

Response: Thank you very much for your kind review and comments. We have replied to every comment made by you, and modified or corrected as much as we could in our manuscript according to your comments. We look forward to your positive consideration for the revised manuscript.

Specific comments:

Comment-1. On the cathode side, it is not clear the function of microorganisms in contact with the cathode material. This requires a detailed explanation.

Response: Thank you for your comment, Line 212-222, Line 227-230, Line 252-259, explained the function of microorganisms in contact with the cathode material. In our research, Xanthobacter, Bradyrhizobium, Mesorhizobium, Achromobacter, Castellaniella, Microbacterium, and Pseudoclavibacter mainly clustered in the cathode, which were responsible for the degradation of SDZ.

Comment-2. -What could be the electron transfer mechanism by microorganisms at the cathode?

Response: Thank you for your comment. In the air-cathode MFCs, there is O2 on the outside of the cathode and microorganisms on the inside, so the electron acceptor is O2 and some cathode microorganisms. The mechanism of O2 as an electron acceptor has been clear that O2 accept electrons and H+ in the solution to generate water [1]. However, the electron transfer mechanism of microorganisms as electron acceptors is still unclear. There have been many researches on the electron transfer pathways of electrogenic microorganisms, including direct electron transfer mediated by cytochrome c membrane proteins or conductive pili and indirect electron transfer mediated by electron shuttle [2]. The microorganisms receive electrons from the electrode is electrotrophic microorganisms [3]. Recently, electrotrophic microorganisms have received more attention, but it is not clear whether electrotrophic microorganisms and electrogenic microorganisms share electron transfer pathways, or whether they have unique electron transfer pathways.

Comment-3. -Transferred electrons from anode to cathode are used to abiotically/biotically reduce O2 into something else?

Response: Thank you for your comment. Transferred electrons from anode to cathode were used to reduce O2 into H2O, and converted nitrate, sulfate to nitrite, hydrogen sulfide [3].

Comment-4.- Were anodic, cathodic or both biofilms subjected to electrochemical analysis?

Response: Thank you for your comment. The electrochemical activity part only compared the effect of the addition of SDZ on the electrochemical activity of MFCs. The anode was mainly responsible for power generation. Therefore, only the anodic biofilms were measured.

Comment-5. -Were biofilms subjected to analysis under the presence of substrate, at the maximum of current production or under complete absence of substrate.

Response: Thank you for your comment. Biofilms with or without substrate were analyzed (Three samples under each condition), but there was no significant difference in microbial community under these two conditions. Therefore, we selected the best parallelism for further analysis. 

References:

  1. Santoro, C., Arbizzani, C., Erable, B., Ieropoulos, I. Microbial fuel cells: From fundamentals to applications. A review. J. Power Sources. 2017, 356, 225-244. https://doi.org/10.1021/es0605016
  2. Logan, B.E., Hamekers B., Rozendal R., Schroder, U., Keller, J., Freguia, S., Aelterman, P., Verstraete, W., Rabaey, K. Microbial fuel cells: methodology and technology. Environ. Sci. Technol. 2018, 40, 5181-5192. http://dx.doi.org/10.1016/j.jpowsour.2017.03.109
  3. Logan, B.E., Rossi, R., Ragab, A., Saikaly, P.E. Electroactive microorganisms in bioelectrochemical systems. Nat. Rev. Microbiol. 2019, 17, 307-319. https://doi.org/10.1038/s41579-019-0173-x

Reviewer 3 Report

The paper of Yang et al. scores high in novelty as microbial fuel cells(MFCs)  are a rather a promising technology which have been insufficiently developed so far. I would like to see a few notes regarding reverse MFCs and how they could be related to antibiotics removal.

The title is representative and the abstract is clear and concise.

Introduction

State the principle of how MFC remove antibiotics, also if there are some antibiotics which can not be removed by the use of MCFs.

State why these species were used.

A detailed description of the air-cathode and the electron exchange should be made.

Materials and methods

Line 97: nature of the material of the 0.45µ membranes.

Line 100: detail the GB/T 1914-1989 method.

Line 102-106: frequency was set at ? Hz. Please add the chromatograms.

Is mutualism between the strains used a factor in the performance of the MFC in the current study? What are the known types of interactions between the strains that were used in the fuel cell. This is an aspect that should be mentioned and clarified .

Author Response

Response to Reviewer:

Reviewer:

The paper of Yang et al. scores high in novelty as microbial fuel cells (MFCs) are a rather a promising technology which have been insufficiently developed so far. I would like to see a few notes regarding reverse MFCs and how they could be related to antibiotics removal.

The title is representative and the abstract is clear and concise.

Introduction

State the principle of how MFC remove antibiotics, also if there are some antibiotics which cannot be removed by the use of MFCs.

State why these species were used.

A detailed description of the air-cathode and the electron exchange should be made.

Materials and methods

Line 97: nature of the material of the 0.45µ membranes.

Line 100: detail the GB/T 1914-1989 method.

Line 102-106: frequency was set at ? Hz. Please add the chromatograms.

Is mutualism between the strains used a factor in the performance of the MFC in the current study? What are the known types of interactions between the strains that were used in the fuel cell. This is an aspect that should be mentioned and clarified.

Response: Thank you very much for your kind review and comments. We have replied to every comment made by you, and modified or corrected as much as we could in our manuscript according to your comments. We look forward to your positive consideration for the revised manuscript.

Specific comments:

Comment-1. Introduction

State the principle of how MFC remove antibiotics, also if there are some antibiotics which cannot be removed by the use of MFCs.

Response: Thank you for your suggestion, we have corrected this part. (Line 40–44). But it is unclear which antibiotics cannot be removed by MFCs.

Comment-2. State why these species were used.

Response: Thank you. We added the reason to choose SDZ. Please check Line 79-80.

Comment-3. A detailed description of the air-cathode and the electron exchange should be made.

Response: Thank you. We have explained the mechanism of MFCs. Please check Line 38-40.

Comment-4. Line 97: nature of the material of the 0.45µ membranes.

Response: Thank you for your comment. We have added filter information (Line 111)

Comment-5. detail the GB/T 1914-1989 method.

Response: Thank you for your comment. We have put the detail method in supplementary materials.

Comment-6. Line 102-106: frequency was set at ? Hz. Please add the chromatograms

Response: Thank you for your comment. Electrochemical impedance spectroscopy was shown in Figure S3b. ESI polt was the fitting graph of experimental results.

Comment-7. Is mutualism between the strains used a factor in the performance of the MFC in the current study? What are the known types of interactions between the strains that were used in the fuel cell. This is an aspect that should be mentioned and clarified.

Response: Thank you for your comment. The mutualism between the strains have been reported in previous researchers [1-2]. In our research, the electrons provided by exoelectrogenic bacteria could increase the metabolic reaction of degradation related bacteria located in the anode and cathode; Degradation related bacteria might contribute to the power generation by producing metabolites, which could be used as electron donors by the electroactive bacteria. (Line 340–343)

References:

  1. Hou, R., Luo, X.S., Liu, C.C., Zhou, L.H., Wen, J.L., Yuan, Y. Enhanced degradation of triphenyl phosphate (TPHP) in bioelectrochemical systems: Kinetics, pathway and degradation mechanisms. Environ. Pollut. 2019, 254, 113040. https://doi.org/10.1016/j.envpol.2019.113040
  2. Zhao, H.H., Kong, C.H. Elimination of pyraclostrobin by simultaneous microbial degradation coupled with the Fenton process in microbial fuel cells and the microbial Community. Bioresour. Technol. 2018, 258, 227-233. https://doi.org/10.1016/j.biortech.2018.03.012

Round 2

Reviewer 3 Report

I find the revision to be good.